# INFINITEMESH: VIEW INTERPOLATION USING MULTI-VIEW DIFFUSION FOR 3D MESH RECONSTRUCTION

## ABSTRACT

We present **InfiniteMesh**, a feed-forward framework for efficient high-quality image-to-3D generation with view interpolation. Recent advancements in Large Reconstruction Model (LRM) have demonstrated significant potential in extracting 3D content from multi-view images produced by 2D diffusion models. Nevertheless, challenges remain as 2D diffusion models often struggle to generate dense images with strong multi-view consistency, and LRMs often exacerbate this multi-view inconsistency during 3D reconstruction. To address these issues, we propose a novel framework based on LRM that employs 2D diffusion-based view interpolation to enhance the quality of the generated mesh. Leveraging multi-view images produced by a 2D diffusion model, our approach introduces an Infinite View Interpolation module to generate interpolated images from main views. Subsequently, we employ a tri-plane-based mesh reconstruction strategy to extract robust tokens from these multiple generated images and produce the final mesh. Extensive experiments indicate that our method generates high-quality 3D content in terms of both texture and geometry, surpassing previous state-of-the-art methods.

## 1 INTRODUCTION

3D generation from a single image has become increasingly vital across various fields, including virtual reality, gaming, and robotics Pang et al. (2024). Recent advancements in 2D diffusion models Ho et al. (2020); Song et al. (2021); Blattmann et al. (2023a) and Large Reconstruction Models (LRMs) Hong et al. (2023); Li et al. (2023); Tang et al. (2024); Wang et al. (2024); Xu et al. (2024a) have opened new avenues for 3D content creation. Several works, such as Poole et al. (2022); Lin et al. (2023); Qian et al. (2023); Seo et al. (2023); Qiu et al. (2024); Chen et al. (2024a;b), leverage 2D diffusion models to generate 3D content through a Score Distillation Sampling (SDS) pipeline. An alternative approach involves creating multi-view images using 2D diffusion, followed by the application of reconstruction algorithms to obtain 3D content from these images Liu et al. (2023a); Shi et al. (2023b); Liu et al. (2023b); Wang & Shi (2023); Shi et al. (2023a); Long et al. (2024).

Nonetheless, current state-of-the-art (SoTA) methods typically produce a limited number of multi-view images (usually four or six), which restricts the generation of geometric and textural details. Approaches such as Blattmann et al. (2023b); Voleti et al. (2024); Chen et al. (2024c) have introduced video diffusion strategies to directly increase the number of generated multi-view images, however, they are often plagued by the challenge of multi-view inconsistency, as illustrated in Fig. 1 (SV3D and V3D). Besides, They also require significant training costs, including GPU memory, etc., which greatly limit their application.

To address these limitations, we introduce **InfiniteMesh**, a novel LRM-based image-to-3D framework, designed to improve 3D generation quality through 2D diffusion-based view interpolation. InfiniteMesh generates a large number of multi-view images with two steps. Firstly, InfiniteMesh employs a 2D diffusion model for $N$ main views generation ($N$ is 4), then, an Infinite View Interpolation (IVI) module is incorporated to generate interpolated images with superior multi-view consistency from main views, enriching representational details. Finally, a tri-plane-based mesh reconstruction model utilizes these views to extract robust tokens, and produce a final mesh that shows high-quality geometry and texture. We validate our approach using the Google Scanned Objects (GSO) dataset Downs et al. (2022) and images collected from the web, demonstrating that InfiniteMesh outperforms existing baseline methods.

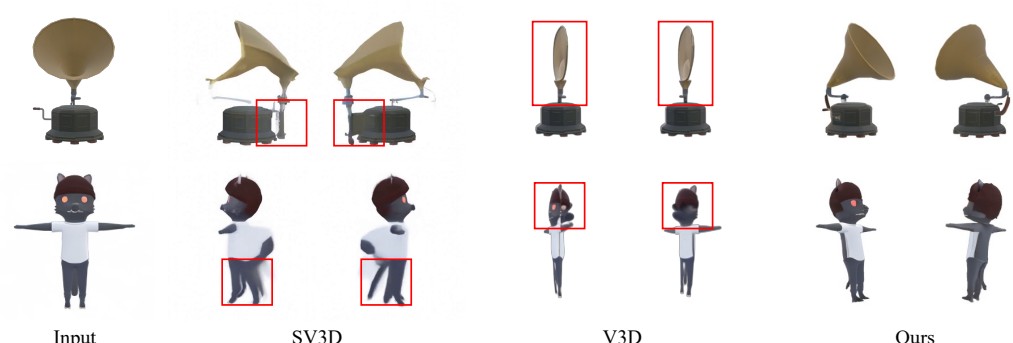

Input          SV3D          V3D          Ours

Figure 1: Qualitative comparisons between our IVI module and video diffusion methods in multi-view generation. Two generated images are shown here, and images generated by video diffusion networks show inconsistencies due to the lack of connectivity across frames. In contrast, our method ensures strong inter-frame connections, which significantly enhances the multi-view consistency of the generated images.

The motivation behind our InfiniteMesh is obvious and straightforward, we separate the process of generating large number of multi-view images into two steps ($N$ main views generation and Infinite View Interpolation (IVI) for view interpolation). IVI module can facilitate consistent image interpolation between two neighbouring main views, better constraints are provided in the view interpolation process, thus better results can be expected. As shown in Fig. 1 (Ours), with such a setting, multi-view consistencies and image qualities can be guaranteed.

Our contributions can be summarized as follows:

- We propose InfiniteMesh, an LRM-based framework to efficiently generate high-quality 3D mesh from a single image, utilizing multi-view diffusion for view interpolation.

- We develop an IVI module that facilitates consistent image interpolation between any two neighbouring main views using 2D multi-view diffusion, followed by a tri-plane-based LRM to enhance mesh texture and geometry.

- We conduct extensive experiments to demonstrate the superiority of our proposed methods over other SoTA methods, both quantitatively and qualitatively.

## 2 RELATED WORKS

### 2.1 3D GENERATION

Recent advancement in diffusion models Sohl-Dickstein et al. (2015) has brought image generation to a new height Ho et al. (2020); Song et al. (2021); Rombach et al. (2022); Blattmann et al. (2023a). Numerous works have focused on leveraging diffusion models for 3D generation. A mainstream approach is directly training 3D generators using 3D ground truth Zhou et al. (2021); Zheng et al. (2023); Wang et al. (2023); Gupta et al. (2023); Shue et al. (2023). For instance, Zhou et al. (2021) and Zheng et al. (2023) trained diffusion models to directly generate 3D voxels. In Wang et al. (2023) and Shue et al. (2023), a 3D-aware tri-plane diffusion model is introduced to produce NeRF Mildenhall et al. (2021) representations. Nonetheless, 3D diffusion methods tend to be time-consuming during optimization, and often show low quality in terms of texture and geometry.

To deal with this, some studies have explored the utilization of 2D diffusion-based generators for 3D generation. DreamFusion Poole et al. (2022) was the first to use 2D diffusion models to generate 3D content through SDS. Building upon this work, Lin et al. (2023); Qian et al. (2023); Seo et al. (2023); Qiu et al. (2024); Chen et al. (2024a;b) have adopted the SDS pipeline to optimize various 3D representations such as NeRF, mesh, and gaussian splatting Kerbl et al. (2023). However, performing 3D generation tasks with 2D diffusion models often encounters issues related to multi-view inconsistency, indicating room for improvement.

## 2.2 MULTI-VIEW DIFFUSION MODELS

Researchers have made great efforts to improve diffusion models in multi-view images generation. Zero123 Liu et al. (2023a) was the first to encode camera pose as an additional condition to generate images from different specific views. On this basis, MVDream Shi et al. (2023b) replace self-attention in the Unet architecture with multi-view attention to facilitate multi-view consistency. Other works Liu et al. (2023b); Wang & Shi (2023); Shi et al. (2023a); Long et al. (2024) share a similar idea to generate 3D-aware and multi-view consistent 2D representations. These multi-view images can be further processed using techniques such as NeRF Mildenhall et al. (2021) and Gaussian Splatting Kerbl et al. (2023) to obtain 3D representations. Nevertheless, existing multi-view diffusion models are constrained to generating a limited number of images from a single input image. Recent advancements Blattmann et al. (2023b); Voleti et al. (2024); Chen et al. (2024c) have sought to outcome this limitation by utilizing temporal priors in video diffusion models to boost the number of generated images. Despite these improvements, such strategies often neglect the connectivity between frames, resulting in inconsistencies and diminishing the quality of the generated 3D content.

## 2.3 LARGE RECONSTRUCTION MODELS

The advent of large-scale 3D datasets Deitke et al. (2023; 2024) has significantly advanced the field of image-to-3D generation, bringing generalized reconstruction models to new heights. LRM Hong et al. (2023) was a pioneer that demonstrates the superiority of Transformer Vaswani et al. (2017) backbone in mapping image tokens to predict tri-plane NeRF under multi-view supervision. Building upon this foundation, Instant3D Li et al. (2023) extends the input to multi-view images, largely enhancing the quality of image-to-3D generation through multi-view diffusion models. Inspired by Instant3D, subsequent methods such as LGM Tang et al. (2024) and GRM Xu et al. (2024b) further refine it by replacing NeRF representations with 3D Gassian Splatting Kerbl et al. (2023) to improve the rendering efficiency. Recently, CRM Wang et al. (2024) and InstantMesh Xu et al. (2024a) take advantage of FlexiCubes Shen et al. (2023) to improve both efficiency and quality of image-to-3D generation.

## 3 INFINITEMESH

As illustrated in Figure 2 (a), given a single input image $x_0$, the architecture of our proposed InfiniteMesh consists of 4 primary components: 1) a multi-view diffusion model to generate main multi-view images, 2) an **I**nfinite **V**iew **I**nterpolation (IVI) module to perform view interpolation between any two neighbouring views, and 3) a tri-plane based large reconstruction model to reconstruct a high-quality 3D mesh. The details of each component are elaborated below.

### 3.1 MULTI-VIEW DIFFUSION MODEL

In this paper, we follow Long et al. (2024) to train a four-view generation model based on multi-view 2D diffusion, which takes a single image as input, and generate outputs from four viewpoints (front, right, back, and left) to maximize multi-view consistency.

### 3.2 INFINITE VIEW INTERPOLATION

Building upon main views generated by the multi-view diffusion model, we perform view interpolation through our IVI module. As depicted in Fig. 2 (b), given two adjacent main view images $x_1^M$ and $x_2^M \in \mathbb{R}^{H \times W \times 3}$, our objective is to learn a model $f$ that synthesizes any interpolated image $x_i$, along with their corresponding camera poses $\Pi = \{\pi_1^M, \pi_i, \pi_2^M\}$. Here $\pi = [R, T]$, where $R \in \mathbb{R}^{3 \times 3}$ and $T \in \mathbb{R}^3$. This relationship can be formulated as follows:

$$x_i = f(x_1^M, x_2^M, \Pi). \tag{1}$$

Most multi-view diffusion architectures Liu et al. (2023a); Long et al. (2024) employ the latent diffusion denoising strategy Rombach et al. (2022). In our view interpolation setting where two main views are input, one view is designated as the reference image $x_i^{Ref}$, and the other as the

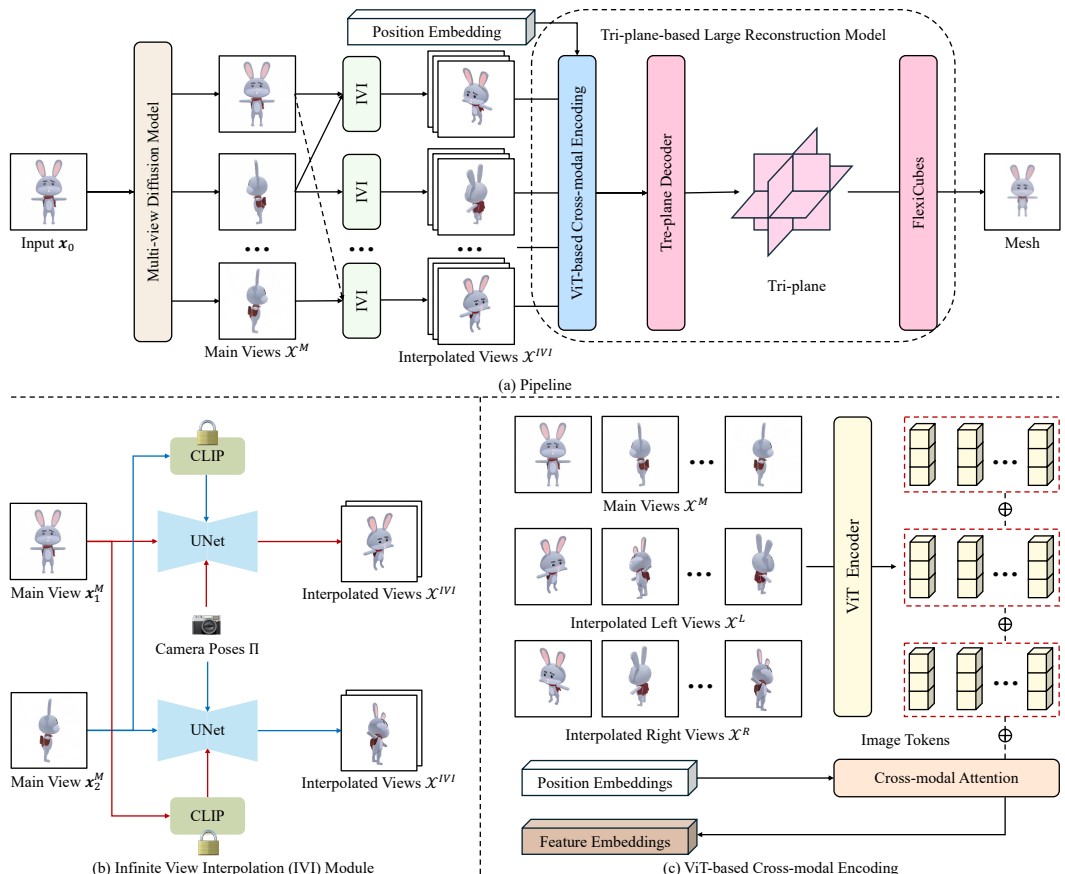

Figure 2: (a) The pipeline of our proposed InfiniteMesh. Starting with a single image, InfiniteMesh first generates main views using a multi-view diffusion model. (b) Interpolated views are then obtained from these main views using IVI module. (c) The images are processed through a ViT to extract feature embeddings, which are then used to generate a high-quality 3D mesh utilizing a tri-plane-based large reconstruction model.

condition image $\boldsymbol{x}_i^{Cond}$, so the adapted objective of the latent diffusion denoising process in our IVI module can be expressed as:

$$L_{IVI} := \mathbb{E}_{\boldsymbol{z} \sim \mathcal{E}(\boldsymbol{x}_i^{Ref}), t, \epsilon \sim \mathcal{N}(0,1)} \left\| \epsilon - \epsilon_\theta(\boldsymbol{z}_t, t, \mathcal{C}(\boldsymbol{x}_i^{Cond}, \boldsymbol{\pi}_i)) \right\|_2^2, \tag{2}$$

where $\mathcal{C}(\boldsymbol{x}_i^{Cond}, \boldsymbol{\pi}_i)$ represents the condition embedding of the condition view and the relative camera pose. The inference model $f$ is optimized to perform iterative denoising from $\boldsymbol{z}_T$ by training the model $\epsilon_\theta$ Rombach et al. (2022). Specifically, $\boldsymbol{z}_T$ is obtained by channel-concatenating $\boldsymbol{x}^{Ref}$. Following Liu et al. (2023a), a CLIP Radford et al. (2021) embedding of $\boldsymbol{x}_i^{Cond}$ is concatenated with $\boldsymbol{\pi}_i$. This ensures that the generated interpolated images maintain multi-view consistency with both $\boldsymbol{x}^{Ref}$ and $\boldsymbol{x}^{Cond}$, which benefits stability of view interpolation.

Given the varying camera poses of each interpolated view, some views are positioned closer to $\boldsymbol{x}_1^M$ while others are nearer to $\boldsymbol{x}_2^M$. To ensure a balanced distribution and multi-view consistency, for $\boldsymbol{x}_i$, the reference and condition views can be expressed as follows:

$$[\boldsymbol{x}_i^{Ref}, \boldsymbol{x}_i^{Cond}] = \begin{cases} [\boldsymbol{x}_1^M, \boldsymbol{x}_2^M], & \text{if } i \leq \dfrac{n}{2}, \\ [\boldsymbol{x}_2^M, \boldsymbol{x}_1^M], & \text{if } i > \dfrac{n}{2}. \end{cases} \tag{3}$$

where n represents the umber of interpolated images. Better constraints are provided in the view interpolation process, thus better results can be expected. In our implementation, we set $n$ to 2, empirically.

In IVI module, two main views are employed as reference and condition to improve the consistency and stability of the interpolated images. The consistent interpolated images effectively supplement missing views, thereby enriching the detail during model reconstruction. We provide more analysis in the experiment section.

### 3.3 TRI-PLANE-BASED MESH RECONSTRUCTION

We train a robust tri-plane-based reconstruction model to obtain high-quality mesh from the multiple generated images. As illustrated in Fig. 2 (c), for every two adjacent main images $\boldsymbol{x}_1^M$ and $\boldsymbol{x}_2^M$, we generate a sequence of interpolated images $\mathcal{X}^{IVI} = \{\boldsymbol{x}_1, \ldots, \boldsymbol{x}_n\}$ through our IVI module. Consequently, for each main view $\boldsymbol{x}_i^M$ in the set of sparse-view main images $\mathcal{X}^M = \{\boldsymbol{x}_1^M, \ldots, \boldsymbol{x}_N^M\}$ that generated by multi-view diffusion model, where $N$ represents the number of main views, we have interpolated images on its left and right: $\mathcal{X}^L = \{\boldsymbol{x}_1^L, \ldots, \boldsymbol{x}_n^L\}$ and $\mathcal{X}^R = \{\boldsymbol{x}_1^R, \ldots, \boldsymbol{x}_n^R\}$, respectively. Following general large reconstruction models Hong et al. (2023); Li et al. (2023); Xu et al. (2024a); Wei et al. (2024); Xu et al. (2024b), we employ a Vision Transformer (ViT) $\mathcal{V}$ Dosovitskiy et al. (2020) to extract image tokens from $\mathcal{X}^M$ and their corresponding $\mathcal{X}^L$ and $\mathcal{X}^R$ and add them to a position embedding through residual connection. This process can be written as follows:

$$\boldsymbol{f}^F = \boldsymbol{p} + \mathcal{A}_{cm}(\boldsymbol{p}, \mathcal{V}(\mathcal{X}^M) \oplus \mathcal{V}(\mathcal{X}^L) \oplus \mathcal{V}(\mathcal{X}^R)), \tag{4}$$

where $\boldsymbol{f}^F$ represents the fused feature embeddings, $\boldsymbol{p}$ represents the initial position embedding, $\oplus$ represents channel-wise concatenation, and $\mathcal{A}_{cm}$ represents a cross-modal attention operation, defined as:

$$\mathcal{A}_{cm}(\boldsymbol{p}, \boldsymbol{f}) = softmax(\frac{\boldsymbol{q}\boldsymbol{k}^T}{\sqrt{d}}) \cdot \boldsymbol{v}, \tag{5}$$

with

$$\boldsymbol{q} = \boldsymbol{w}_q \cdot \boldsymbol{p}, \quad \boldsymbol{k} = \boldsymbol{w}_k \cdot \boldsymbol{f}, \quad \boldsymbol{v} = \boldsymbol{w}_v \cdot \boldsymbol{f}, \tag{6}$$

where $\boldsymbol{w}$ denotes learnable projection matrices Vaswani et al. (2017); Dosovitskiy et al. (2020). In this learnable way, the main and interpolated image tokens are fused via residual connection to enhance multi-view consistency. Subsequently, following InstantMesh Xu et al. (2024a), we decode $\boldsymbol{f}^F$ to obtain a tri-plane representation, and reconstruct the final mesh through FlexiCubes Shen et al. (2023). Thanks to our IVI module, more multi-view consistent image tokens are provided, bringing more details related to texture and geometry, thus resulting in a high-quality reconstructed mesh.

The loss function for mesh reconstruction can be expressed as follows:

$$\begin{aligned} \mathcal{L} = &\mathcal{L}_{rgb} + \lambda_{lpips}\mathcal{L}_{lpips} + \lambda_{mask}\mathcal{L}_{mask} \\ &+ \lambda_{depth}\mathcal{L}_{depth} + \lambda_{normal}\mathcal{L}_{normal} + \lambda_{reg}\mathcal{L}_{reg}, \end{aligned} \tag{7}$$

with $\lambda_{lpips} = 2.0$, $\lambda_{mask} = 1.0$, $\lambda_{depth} = 0.5$, $\lambda_{normal} = 0.2$, $\lambda_{reg} = 0.01$. Readers may refer to Xu et al. (2024a) for more details. During training of mesh reconstruction, we randomly select 4 views as supervision.

## 4 EXPERIMENTS

In this section, we conduct a series of experiments quantitatively and qualitatively to evaluate the performance of our proposed InfiniteMesh. We compare InfiniteMesh against SoTA multi-view and image-to-3D baseline methods. Additionally, we perform ablation studies to validate the effectiveness and expand-ability of our proposed IVI module.

### 4.1 EXPERIMENTAL SETTINGS

**Dataset.** Following prior research Liu et al. (2023a;b); Long et al. (2024), we utilize the Google Scanned Objects dataset Downs et al. (2022) for our evaluation, which encompasses a diverse array of common everyday objects. For the evaluation phase, we choose 30 representative objects ranging from everyday items to animals. Besides, images collected from web are also evaluated to prove our robustness.

Table 1: Quantitative comparison for geometry quality between our method and baselines for 3D textured mesh generation. We report Chamfer Distance, Volume IoU and F-score on the GSO dataset. The best results are shown in bold font.

| Method | Chamfer Dist. ↓ | Vol. IoU ↑ | F-Sco. ↑ |
|---|---|---|---|
| One-2-3-45 | 0.0172 | 0.4463 | 0.7219 |
| SyncDreamer | 0.0140 | 0.3900 | 0.7574 |
| Wonder3D | 0.0186 | 0.4398 | 0.7675 |
| Magic123 | 0.0188 | 0.3714 | 0.6066 |
| LGM | 0.0117 | 0.4685 | 0.6869 |
| InstantMesh | 0.0103 | 0.5712 | 0.7121 |
| V3D | 0.0143 | 0.4660 | 0.6234 |
| SV3D | 0.0142 | 0.4949 | 0.6529 |
| **Ours** | **0.0101** | **0.6399** | **0.7765** |

**Implementation Details.** Our model is trained on the LVIS subset of the Objaverse dataset Deitke et al. (2023), consisting of approximately 30,000+ objects after a thorough cleanup process. For image interpolation, we fine-tune our IVI module starting from Wonder3D Long et al. (2024), which has previously been fine-tuned for multi-view generation. During the fine-tuning process, we resize the image to $256 \times 256$ and employ a batch size of 128. This fine-tuning is performed for 10,000 steps. For mesh reconstruction, starting from InstantMesh Xu et al. (2024a), we fine-tune the model for 30,000 steps with a total batch size of 4. We use eight Nvidia A100 40GB in this paper. In both fine-tuning processes, we remain the original optimizer settings and $\epsilon$-prediction strategy.

**Baselines and Metrics.** For comparative analysis, we adopt One-2-3-45 Liu et al. (2024), SyncDreamer Liu et al. (2023b), Wonder3D Long et al. (2024), Magic123 Qian et al. (2023), LGM Tang et al. (2024), InstantMesh Xu et al. (2024a), V3D Chen et al. (2024c), and SV3D Voleti et al. (2024) as our baselines to evaluate the quality of the generated mesh. We also adopt V3D and SV3D to evaluate the quality of novel view synthesis of our IVI module in orbiting view generation.

To evaluate the geometry quality for 3D textured mesh generation, Chamfer Distances, Volume IoU, and F-score metrics are utilized. To evaluate novel view synthesis (NVS) and the texture quality for 3D texutred mesh generation, we employ the PSNR, SSIM Wang et al. (2004), and LPIPS Zhang et al. (2018) metrics. We also evaluate the GPU memory usage in orbiting view generation.

## 4.2 3D Textured Mesh Generation

The quantitative results are summarized in Tabs. 1 and 2, where our InfiniteMesh outperforms all baseline methods in terms of both geometric and texture quality metrics. For mesh texture evaluation, we render 24 images at $512 \times 512$ resolution, capturing meshes at elevation angles of $0°$, $15°$, and $30°$, with 8 images evenly distributed around a full $360°$ rotation for both generated and ground-truth meshes. Among the baseline models, though InstantMesh demonstrates better performance in geometry quality, and SV3D demonstrates better performance in texture quality, our results outperform these SOTAs in both geometry and texture. Based on high-quality main view results, the diverse detail acquisition from the IVI module enables the reconstruction model to capture comprehensive geometric and texture information, which is proved in ablation studies in Sec. 4.4.

Qualitative comparisons in Fig. 3 including images collected from web and the GSO dataset. Our consistent view interpolation approach enriches image tokens within the reconstruction model, providing more features with good multi-view consistency, therefore, comparing with SOTAs, more smooth geometry and visual appealing textures can be obtained by our approach.

## 4.3 Novel View Synthesis

We benchmark the novel view synthesis capabilities of our IVI module against video diffusion-based baselines in orbiting view generation, where 12 views are selected along a horizontal orbiting trajectory. Quantitative results are presented in Tab. 3. Our approach effectively employ two main views as reference and condition, thus improving the consistency and stability of the interpolated images. As shown in Tab. 3, it is also worth mentioning that our IVI module requires a much lower memory cost for inference compared to video diffusion-based methods, as we generate views by two steps.

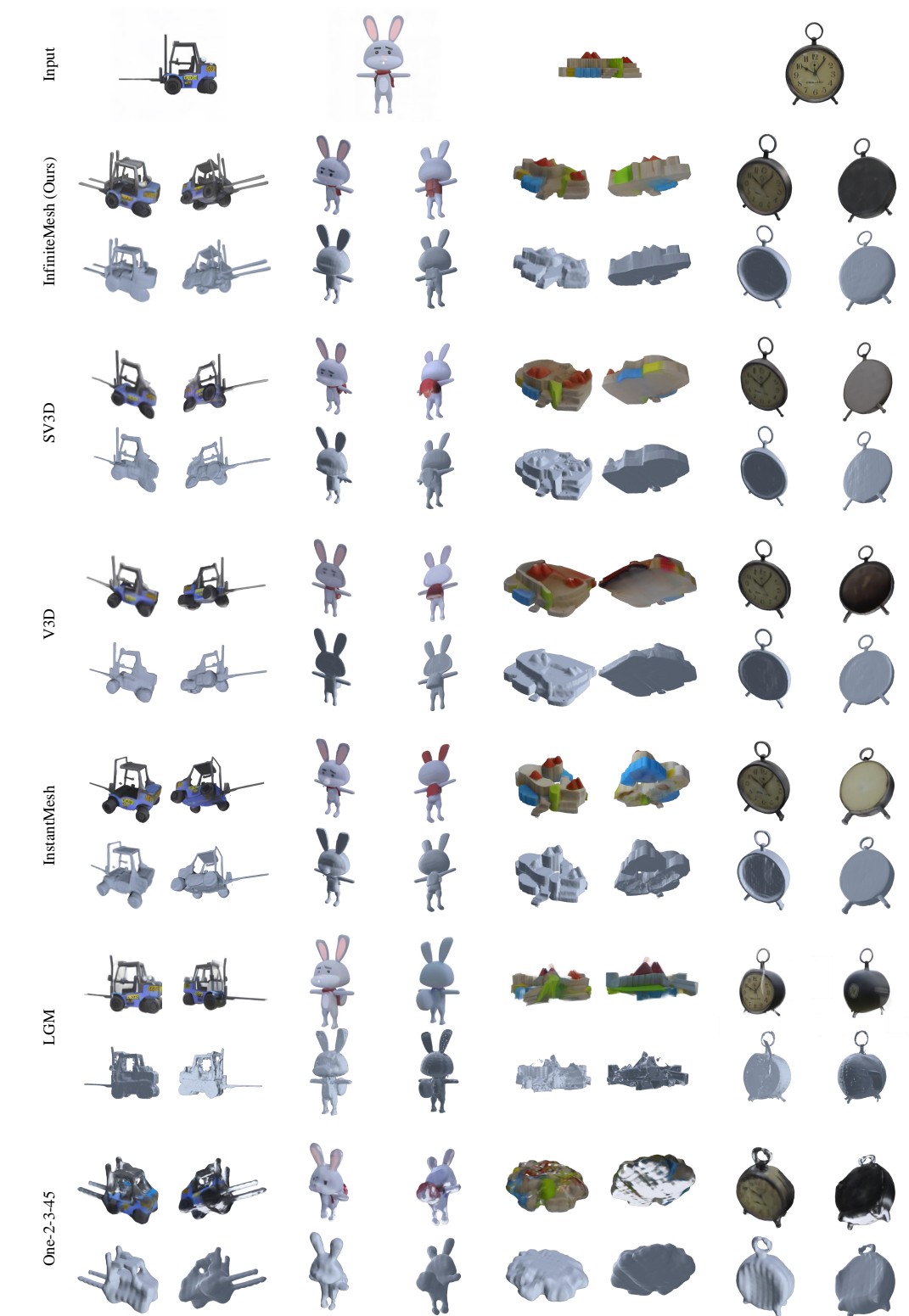

Figure 3: Qualitative 3D mesh results generated by InfiniteMesh demonstrate better geometry and texture compared to other baselines.

Table 2: Quantitative comparison for texture quality between our method and baselines for 3D textured mesh generation. We report PSNR, SSIM Wang et al. (2004), LPIPS Zhang et al. (2018) on the GSO dataset. The best results are shown in bold font.

| Method | PSNR ↑ | SSIM ↑ | LPIPS ↓ |
|---|---|---|---|
| One-2-3-45 | 13.93 | 0.8084 | 0.2625 |
| SyncDreamer | 14.00 | 0.8165 | 0.2591 |
| Wonder3D | 13.31 | 0.8121 | 0.2554 |
| Magic123 | 12.69 | 0.7984 | 0.2442 |
| LGM | 13.28 | 0.7946 | 0.2560 |
| InstantMesh | 17.66 | 0.8053 | 0.1517 |
| V3D | 17.60 | 0.8115 | 0.1520 |
| SV3D | 17.76 | 0.8173 | 0.1517 |
| **Ours** | **18.32** | **0.8230** | **0.1397** |

Table 3: Quantitative comparison between our method and video diffusion-based methods for novel view synthesis in orbiting view generation. We select 12 views along a horizontal orbiting trajectory and report PSNR, SSIM Wang et al. (2004), LPIPS Zhang et al. (2018), GPU memory usage on the GSO dataset. The best results are shown in bold font.

| Method | PSNR ↑ | SSIM ↑ | LPIPS ↓ | Memory(MiB) ↓ |
|---|---|---|---|---|
| V3D | 16.37 | 0.796 | 0.173 | 39786 |
| SV3D | 17.12 | 0.801 | 0.185 | 39014 |
| **Ours** | **17.38** | **0.803** | **0.159** | **9686** |

Figure 4: IVI results of elevated camera trajectories and their corresponding reconstructed meshes. To highlight the differences, we present the results with and without a $30°$ elevation.

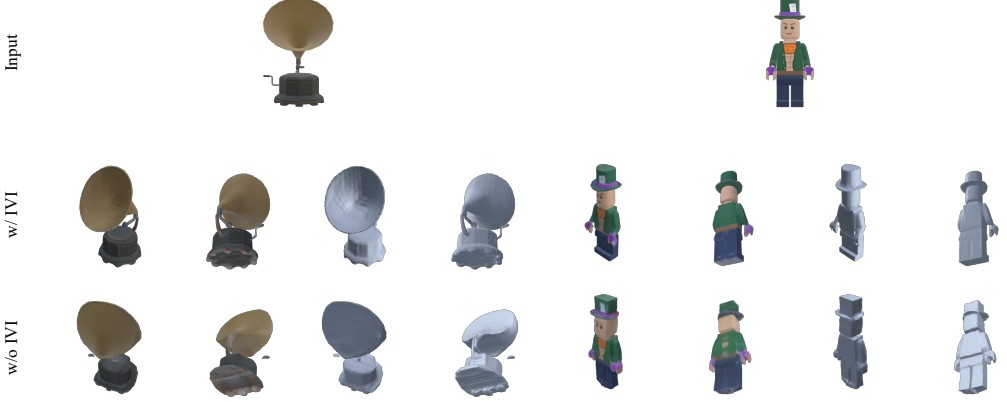

Figure 5: We validate the effectiveness of our IVI module. It can be observed that view interpolation demonstrate better geometry and texture with more details.

## 4.4 ABLATION STUDY

In this subsection, we conduct ablation study to validate the superiority of our architecture.

Table 4: Quantitative results for texture and geometry quality of our method with different elevation angles for 3D textured mesh generation. We report Chamfer Distance, Volume IoU, F-score, PSNR, SSIM Wang et al. (2004), LPIPS Zhang et al. (2018) on the GSO dataset. The best results are shown in bold font.

| Method | Chamfer Dist. ↓ | Vol. IoU ↑ | F-Sco. ↑ | PSNR ↑ | SSIM ↑ | LPIPS ↓ |
|---|---|---|---|---|---|---|
| baseline w/o IVI | 0.0186 | 0.4398 | 0.7675 | 13.31 | 0.8121 | 0.2554 |
| w/o elev. | 0.0102 | 0.6299 | 0.7686 | 18.19 | 0.8222 | 0.1417 |
| w/ +15° and -15° elev. | 0.0101 | 0.6380 | 0.7753 | **18.32** | **0.8230** | 0.1399 |
| w/ +30° and -15° elev. | **0.0101** | 0.6353 | 0.7734 | 18.27 | 0.8229 | **0.1397** |
| w/ +30° and -30° elev. | 0.0101 | **0.6399** | **0.7765** | 18.28 | 0.8229 | 0.1405 |

Table 5: Quantitative results for texture and geometry quality of our method with different number of interpolated number $n$ for 3D textured mesh generation. We report Chamfer Distance, Volume IoU, F-score, PSNR, SSIM Wang et al. (2004), LPIPS Zhang et al. (2018) on the GSO dataset. The best results are shown in bold font.

| $n$ | Chamfer Dist. ↓ | Vol. IoU ↑ | F-Sco. ↑ | PSNR ↑ | SSIM ↑ | LPIPS ↓ |
|---|---|---|---|---|---|---|
| 1 | 0.0101 | 0.6297 | 0.7683 | 18.16 | 0.8209 | 0.1430 |
| 2 | 0.0102 | **0.6380** | **0.7753** | 18.19 | **0.8222** | **0.1417** |
| 3 | **0.0101** | 0.6340 | 0.7719 | **18.21** | 0.8221 | 0.1424 |

**View interpolation for LRM:** To evaluate the effectiveness of view interpolation in our LRM framework, we conduct ablation study with four views (front, right, back, and left) as input and tri-plane-based LRM for reconstruction. As illustrated in Fig. 5, with the IVI module generating interpolated images with superior multi-view consistency, our InfiniteMesh reconstructs high quality meshes with more details and less breakage regarding geometry and texture, especially for objects with complicated geometry and texture. Meanwhile, as shown in Tab. 4, the baseline results are obtained with wonder3D since we use it as baseline without using IVI module. As shown in Tab. 4, results with our IVI module with and without elevation all outperform baseline with large margins, which proves that all our designed camera trajectories work positively for dense image generation.

**Camera pose trajectories in IVI:** Tab. 4 illustrates the impact of varying elevation angles on camera pose trajectories within the IVI module, with representative examples provided in Fig. 4. It can be observed that incorporating elevated camera trajectories (from $\pm 15°$ to $\pm 30°$) within the IVI module show improvements in both geometry and texture. This improvement is attributed to the richer detail diversity provided by elevated camera angles, as evidenced in the 3rd column in Fig. 4.

**Number of interpolation views:** We performed ablation studies to determine the optimal number $n$ of interpolated views. As illustrated in Table 5, with setting $n = 2$ yields the better performance in terms of both geometry and texture quality. Notably, when $n$ is set to 3, silimar results can be obtained comparing with $n = 2$. Therefore, we set $n = 2$ in our experiment.

## 5 LIMITATION AND CONCLUSION

In this paper, we introduce InfiniteMesh, a novel LRM-based image-to-3D framework to produce high-quality 3D content. Particularly, we propose an innovative multi-view diffusion-based IVI module to perform view interpolation, followed by a tri-plane-based mesh reconstruction to obtain the final mesh. Our experimental results indicate the superior performance of InfiniteMesh, demonstrating its ability to generate 3D meshes with exceptional texture and geometric fidelity, compared to existing SoTA methods.

Based on our view interpolation strategy, we can achieve further view expansion of diverse trajectories by further applying the IVI module between the generated images. However, the performance of IVI module depends on the generation qualities of main view images in the first step. We believe improvements can be made by incorporating view super-resolution concept into multi-view diffusion at the feature level, which will be a primary focus of our future work.

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

Table 6: Inference time comparisons between our approach and SOTA video generation methods.

| Methods | inference time (s) |
|---|---|
| SV3D | 85.198 |
| V3D | 31.893 |
| IVI (Ours) | 14.324 |

## A  INFERENCE TIME

**Mesh reconstruction**: Our 3D mesh reconstruction LRM part takes an average time of 1.464 seconds for inference, which is similar with InstantMesh that constructs meshes in an average time of 1.270 seconds.

As shown in Figure 2 (c) and Equation 4 of our main paper, all image tokens are concatenated for subsequent operations. We have a position embedding $p \in \mathbb{R}^{V,P,D}$ and a concatenated tensor $\mathcal{X} \in \mathbb{R}^{V,P,D}$, where $V$ represents the view number. $p$ serves as the query and $\mathcal{X}$ acts as the key in the cross-modal attention operation.

Please kindly note that our approach does not result in a computational time proportional to $V^2$. This is because we only increase the computational load in the image encoder's transformer (cross-modal attention) part. After this step, we employ a Triplane transformer that concatenates and flattens features from all views, then decodes them into a fixed-shape Triplane. Subsequent operations are based on this fixed-shape Triplane, which does not increase computational overhead. Therefore, the additional computational time is primarily confined to the image encoder section, and the overall computational complexity is not proportional to $V^2$.

Besides, as we described before, for the concatenated tensor $X \in \mathbb{R}^{V,P,D}$, though the theoretical time complexity of cross attention is $O((VP)^2, D)$, we use Pytorch **?** in our experiments, the matrix multiplication is mainly performed along P and D dimensions, and "FlashAttention-2: Faster Attention with Better Parallelism and Work Partitioning" and "Memory-Efficient Attention" are utilized to accelerate the attention process. Thus increase of $V$ bring acceptable time consuming, from $1.270$ seconds to $1.464$ seconds.

**View Interpolation**: Tab. 6 demonstrates the inference time comparisons between our approach and video generation methods. Our IVI module takes 3.5s for a single view interpolation process. In our experiment, four interpolations are required, the total video generation time is approximately 14s. The quantitative comparison results with SOTA video generation methods are as follows:

Please kindly note that all results are obtained with a A40 GPU.

## B  VIDEO AND MESH RESULTS ON OOD DATA

We provide more out-of-distribution (OOD) visual results in Fig. 6 with different images as input, including both video and mesh results. We choose images from real-world, Objaverse dataset, and web (both artistic and photographic style), and our model is only trained with Objaverse dataset, which proves the generalization abality of our approach.

As shown in the video results in Fig. 6, better multi-view consistency images can be obtained by our approach, compared with other video-based methods, and differences in the mesh results are highlighted in red areas. For example, our method outperforms other video-based methods with more accurate geometry details in the forklift and cat, while SV3D and V3D show flattened results, treating three-dimensional objects as nearly two-dimensional objects. In the milk case, our approach effectively converts 2D artistic images into consistent multi-view images and intact meshes, maintaining shape consistency that others fail to achieve. Additionally, our method reconstructs more consistent details in the doll's arm, as highlighted in red areas, while other video-based methods result in texture blurring issue.

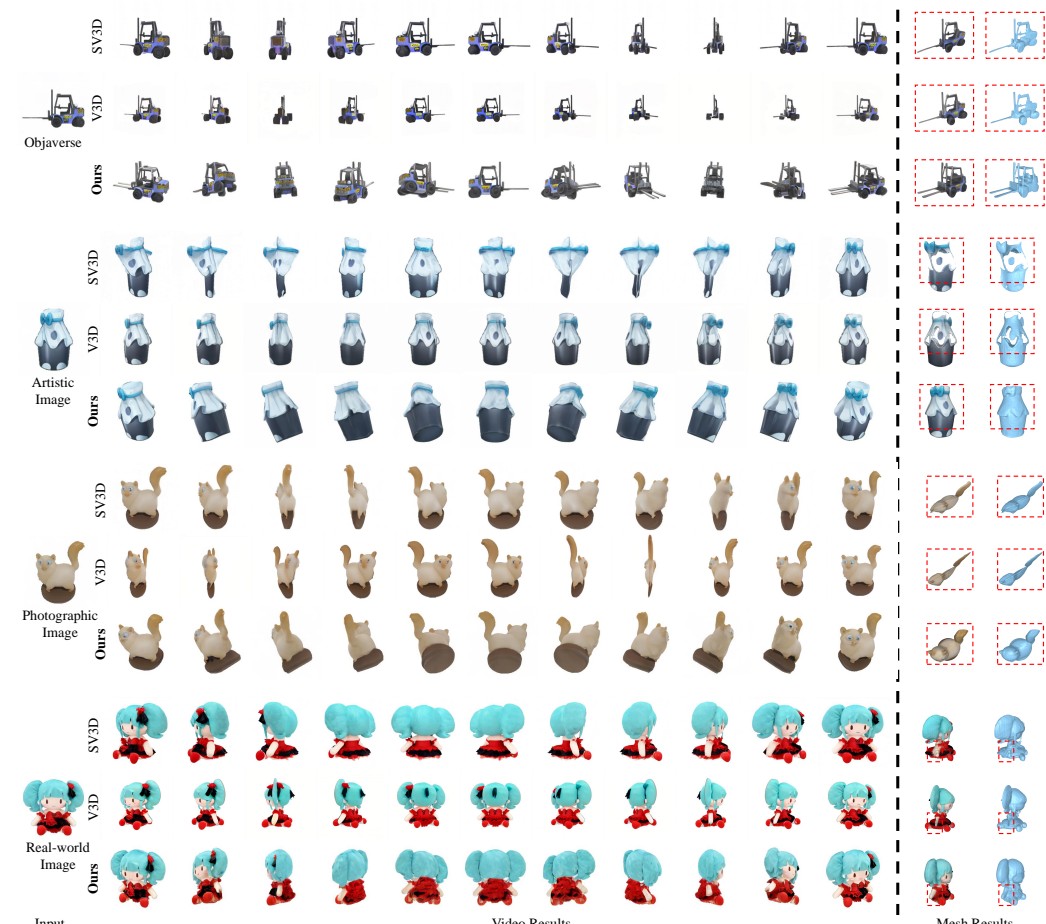

Figure 6: Video and mesh results on out-of-distribution (OOD) data.

## C    QUALITATIVE RESULTS ON CAMERA TRAJECTORIES

We present more distinctive qualitative results on camera trajectories in Fig. 7. We highlight the differences in red areas in the final mesh geometry. With elevation in camera trajectories, our IVI module shows better quality in the reconstructed mesh. For example, the fork of the forklift and the eyes of the dragon are more complete and refined.

## D    360° RECONSTRUCTION DENSE IMAGES

We also present rendered 360° reconstruction dense images to better show the details of our mesh results, as shown in Fig. 7.

## E    LOSS FUNCTION FOR MESH RECONSTRUCTION

The loss function for mesh reconstruction can be expressed as follows:

$$\mathcal{L} = \mathcal{L}_{rgb} + \lambda_{depth}\mathcal{L}_{depth} + \lambda_{normal}\mathcal{L}_{normal}$$
$$+ \lambda_{mask}\mathcal{L}_{mask} + \lambda_{lpips}\mathcal{L}_{lpips} + \lambda_{reg}\mathcal{L}_{reg}, \tag{8}$$

where $\mathcal{L}_{rgb}$, $\mathcal{L}_{depth}$, $\mathcal{L}_{normal}$, and $\mathcal{L}_{mask}$ refer to the loss of RGB images, depth, normal, and mask maps of the reconstructed mesh, and $\mathcal{L}_{lpips}$ and $\mathcal{L}_{reg}$ refer to LPIPS Zhang et al. (2018) and regression loss, respectively, with $\lambda_{lpips} = 2.0$, $\lambda_{mask} = 1.0$, $\lambda_{depth} = 0.5$, $\lambda_{normal} = 0.2$, $\lambda_{reg} = 0.01$. Readers may refer to Xu et al. (2024a) for more details.

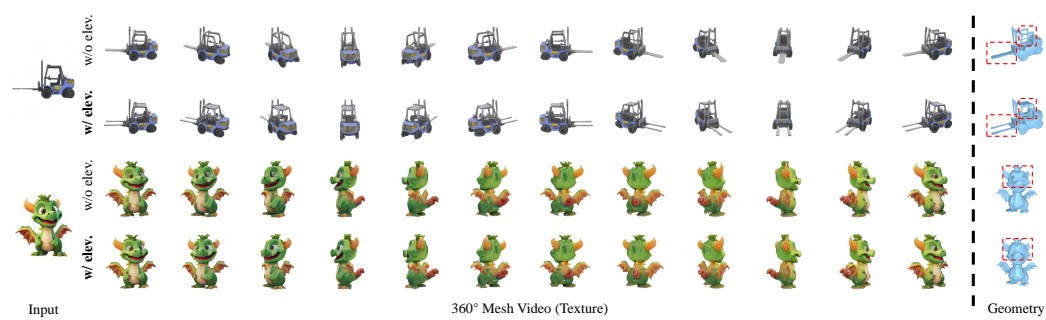

Figure 7: Qualitative results on camera trajectories and 360° reconstruction dense images.

## F  EXPERIMENTAL SETTINGS

For comparative analysis, we adopt One-2-3-45 Liu et al. (2024), SyncDreamer Liu et al. (2023b), Wonder3D Long et al. (2024), Magic123 Qian et al. (2023), LGM Tang et al. (2024), InstantMesh Xu et al. (2024a), V3D Chen et al. (2024c), and SV3D Voleti et al. (2024) as our baselines to evaluate the quality of the generated mesh. We also adopt V3D and SV3D as our baselines to evaluate the quality of novel view synthesis of our IVI module in orbiting view generation.

Please kindly note that we follow the commonly accepted settings and baselines, for example, LGM's performance is compared in both the V3D Chen et al. (2024c) and InstantMesh Xu et al. (2024a).

On the other hand, LGM and other baselines are all methods for 3D generation, though with different technical approaches, making the comparison reasonable.

