# OpenReview forum: "InfiniteMesh: View Interpolation using Multi-view Diffusion for 3D Mesh Reconstruction"
_ICLR.cc/2025/Conference — Submitted to ICLR 2025_

### Official Review · Reviewer_Ystn · 2024-11-01

**Soundness:** 3
**Presentation:** 3
**Contribution:** 2
**Rating:** 3
**Confidence:** 4

**Summary:**

This paper proposed an LRM-based image-to-3D generation with view interpolation. The proposed method consists of two key components: (1) an interpolation model that effectively expands 4 input views to 16 views by synthesizing intermediate viewpoints from multiview diffusion model outputs, and (2) a final LRM model that processes these interpolated views for 3D reconstruction. Experimental results demonstrate that this approach generates high-quality 3D content with superior texture and geometry, outperforming previous state-of-the-art methods.

**Strengths:**

1. The paper's idea is easy to follow.
2. The proposed method demonstrates improvements in both accuracy and computational efficiency while enhancing the continuity and quality of the generated 3D contents.

**Weaknesses:**

1. The proposed method is merely a combination of existing methods, lacking novelty.
2. The model lacks evaluation of OOD (out-of-distribution) datasets or real-world datasets, making its generalization capability questionable.
3. The experimental settings are unclear in the comparative experiments. LGM is a model that converts 4 views to 3D Gaussians, yet it appears in Table 2 in single-image-to-3D comparison experiments. The authors need to further clarify the evaluation details of other models.

**Questions:**

1. When training the multi-view enhanced InstantMesh, are the input multiple views derived from the previous interpolation step or directly using ground truth ？
2. Have you considered performing secondary interpolation on the already interpolated views to obtain more viewpoint data to further improving the model's performance?

---

> ### Author Response · Authors · 2024-11-20
>
> Thank you for finding our paper's ideas easy to follow and recognizing the improvements in accuracy, computational efficiency, and the enhanced continuity and quality of the generated 3D content. Your feedback is greatly appreciated and encourages us to continue our work in this area.
>
> Please kindly note that we update more experiment results and analysis in our updated paper, please re-download our main paper and supplementary materials
>
> Q1: Combination of existing methods, lacking novelty
>
> A1: We respectfully disagree. The proposed IVI module is definitly different with previous SOTA approaches.
>
> First, as mentioned in lines 54-77 and 216-219 of our main paper, the main motivation of our paper is that simply adding dense views does not ensure better reconstruction results, but requires inter-frame consistency. We propose to solve this issue by introducing view interpolation (IVI module) instead of traditional video-based generation to obtain consistent dense views for 3D reconstruction, which is definitely different with previous SOTA approaches.
>
> As shown in Table. 2, Figure. 3 of our main paper, Table. 7, Figure. 6 and Figure. 7 of the Appendix in our updated paper, where our IVI module shows strong multi-view consistency in the generated video compared with video-based dense view generation methods.
>
>
>
> Q2: Evaluation of OOD (out-of-distribution) datasets or real-world datasets.
>
> A2: As mentioned in lines 264-269 and 285-286, we train our model on the synthetic Objaverse dataset and and evaluate on the real-world GSO dataset, so the setting is both OOD and real-world.
>
> Meanwhile, we have included complex results with different input types in Figure. 6 and Figure. 7 in the Appendix of our updated paper, such as from Objaverse image, Artistic image, Photographic image and Real-world captured image. As shown in Figure. 6, as highlighted in red areas, compared with SV3D and V3D, better geometry consistency can be obtained by our method.
>
> As shown in the results in Figure. 6 of the Appendix, better multi-view consistency images can be obtained by our approach, compared with other video-based methods, and differences in the mesh results are highlighted in red areas. For example, our method outperforms other video-based methods with more accurate geometry details in the forklift and cat, while SV3D and V3D show flattened results, treating three-dimensional objects as nearly two-dimensional objects. In the milk case, our approach effectively converts 2D artistic images into consistent multi-view images and intact meshes, maintaining shape consistency that others fail to achieve. Additionally, our method reconstructs more consistent details in the doll's arm, as highlighted in red areas, while other video-based methods result in texture blurring issue.
>
>
>
> Q3: Experimental settings in the comparative experiments, and comparisons with LGM.
>
> A3:  As mentioned in lines 293-303, we followed the commonly accepted settings and baselines. LGM's performance is compared in both V3D and InstantMesh.
>
> On the other hand, LGM generates multi-view images with text or single image as input for 3D generation, though with different technical approaches, making the comparison reasonable. We add more explanations in the appendix-section. G in our updated paper.
>
> [1] Chen Z, Wang Y, Wang F, et al. V3d: Video diffusion models are effective 3d generators[J]. arXiv preprint arXiv:2403.06738, 2024.
> [2] Xu J, Cheng W, Gao Y, et al. Instantmesh: Efficient 3d mesh generation from a single image with sparse-view large reconstruction models[J]. arXiv preprint arXiv:2404.07191, 2024.
>
>
>
> Q4: When training the multi-view enhanced InstantMesh, are the input multiple views derived from the previous interpolation step or directly using ground truth？
>
> A4: Thank you for bringing this up. During the training of the LRM component, we use the ground truths, and we use the predicted images and dense multi-view images obtained from IVI module for mesh reconstruction and texture generation. We will include this clarification in our revised paper.
>
>
>
> Q5:  Performances of secondary interpolation on the already interpolated views
>
> A5: We do consider this setting, however, as mentioned in lines 469-472 and Table. 5, we present a quantitative analysis on the interpolation number n, proving that n=2 yields the better results. Since further increasing n leads to a limited performance increasing in reconstruction results, which prove that adding too many views cannot improve the model's performance.

---

> > ### Comment · Reviewer_Ystn · 2024-11-27
> >
> > Regarding Table 2, there are significant discrepancies in the PSNR values reported for several methods (SyncDreamer, Wonder3D, Magic123, and LGM) compared to their original papers.
> > In particular, I noticed that  Figure 3, LGM showing lower elevation angle than other methods, which appears inconsistent with the results reported in its original paper.
> > This inconsistency raises concerns about the alignment of evaluation settings across different methods. I would recommend:
> > - Clarifying the exact evaluation protocol used, including:
> > - Camera viewpoint settings
> > - Rendering parameters
> >
> > I think that this work essentially presents a simple combination of existing methods, where the proposed IVI module appears more as a technical trick rather than a methodological breakthrough. I fail to see valuable insights into modern 3D DIT (Diffusion Transformer) pipeline or other advanced 3D generation methods.
> > Given ICLR's standards for novel contributions and methodological innovations, I do not believe this paper meets the acceptance criteria for the conference.

---

> > > ### Author Response · Authors · 2024-11-27
> > >
> > > We respectfully disagree with your judgement.
> > >
> > > After reading the review comments, we find other reviewers acknowledge our contributions with (1). Reviewer 8fJs: The IVI module is a novel contribution; (2). Reviewer D6WW: The paper compares the recent state-of-the-art image to 3D models, which is good; (3). Reviewer he5a: Consistency in novel view synthesis is at the core of many image-to-3D task.
> > >
> > > 1.	Our approach is definitly different with previous SOTA approaches, which is effective for 3D generation. First, as mentioned in lines 54-77 and 216-219 of our main paper, the main motivation of our paper is that simply adding dense views does not ensure better reconstruction results, but requires inter-frame consistency. We propose to solve this issue by introducing view interpolation (IVI module) instead of traditional video-based generation to obtain consistent dense views for 3D reconstruction, which is definitely different with previous SOTA approaches. Besides, our approach is totally different with 3D DIT, and we provide various types of results to prove the strong multi-view consistency ability of our IVI module.
> > >
> > > 2. To compare equally, for mesh texture evaluation in Table. 2, we render 24 images at 512*512 resolution, capturing meshes at elevation angles of 0°, 15°, and 30°, with 8 images evenly distributed around a full 360° rotation for both generated and ground-truth meshes. This setting is different from those in SyncDreamer, Wonder3D, Magic123, and LGM. Thus, the PSNR values are different with their original paper.
> > >
> > > 3. Our results outperform LGM with better geometry and textures, which are really easy and obvious to find. We will modify the difference in the final version of our paper.

---

### Official Review · Reviewer_he5a · 2024-11-03

**Soundness:** 3
**Presentation:** 3
**Contribution:** 2
**Rating:** 5
**Confidence:** 3

**Summary:**

This paper presents an effcient image-to-3d generation method with improved multi-view consistency. Building on LRM and 2d diffusion model, it introduces an Infinite View Interpolation (IVI) module, designed to create two interpolated images between main views to improve the quality and consistency of the generated 3D mesh. A tri-plane-based mesh reconstruction is then used to convert these multi-view images into a final 3D mesh with superior texture and geometry. The method demonstrates great generation results.

**Strengths:**

1. The paper is well-written and easy to follow. The author has effectively organized the content and presented ideas in a clear and coherent manner.
2. Consistency in novel view synthesis is at the core of many image-to-3D task. The proposed method is able to outperform V3D and SV3D in multi-view consistency by a large margin. It seems that the introduced IVI module is efficient for generating more consistent meshes.
3. The paper demonstrates its effectiveness through substantial qualitative and quantitative experiments to verify its method outperforms baselines.

**Weaknesses:**

1. The framework primarily builds on existing methods, incorporating previously developed components like the multi-view diffusion model Wonder3D and tri-plane-based Large Reconstruction Model (LRM). I expect the performance gain is great. However, this work does not show video results, and the results in Figure 3 is only marginally better than baselines. It is hard to validate the performance of this work.
2. simple comparison cases: The experimental comparisons mainly focus on simple cases. Including more complex and diverse cases would better showcase the model's strengths and highlight its potential advantages over competing methods.

I am open to further dicussion and willing to raise the score if the concerns are solved.

**Questions:**

As shown in the experiments, InfiniteMesh generates 3D textured meshes based on unnatural images. In contrast, if natural images are used, will the results still maintain good geometry consistency? Additionally, how does the proposed method perform in more complex cases?
It would be beneficial to present some ablation studies on the IVI module, particularly examining the design of separated reference and condition image.

---

> ### Author Response · Authors · 2024-11-20
>
> Thank you for finding our paper well-written, our IVI module efficient in enhancing multi-view consistency, and that our method's effectiveness was evident through our experiments. Your acknowledgment of our work's strengths is greatly appreciated.
>
> Please kindly note that we update more experiment results and analysis in our  updated paper, please re-download our paper for more details.
>
>
> Q1: More dense multi-views results
>
> A1: Thank you for this valuable suggestion. We provide more dense multi-view images results in Figure. 6 in Appendix of our updated paper with different types of input. We also provide more 360° reconstruction dense images in Figure. 7 of the Appendix to better show the details of our reconstructed mesh.
>
> According to Figure. 6 and Figure. 7, our method outperforms SV3D and V3D with better mesh and texture reconstruction on different types of input, such as such as images from real-world, etc. which prove the generalization abalities of our method. Please kindly note that our model is only trained with objaverse dataset. Please see Figure.6 and Figure. 7 of the Appendix for more details.
>
>
>
> Q2: Performance on more complex and diverse scenarios
>
> A2: Thank you for this feedback. In Figure. 6 of the Appendix in our updated paper, we provide results with more complex and diverse scenarios, and differences in the mesh results are highlighted in red areas. For example, our method outperforms other video-based methods with more accurate geometry details in the forklift and cat, while SV3D and V3D show flattened results, treating three-dimensional objects as nearly two-dimensional objects. In the milk case, our approach effectively converts 2D artistic images into consistent multi-view images and intact meshes, maintaining shape consistency that others fail to achieve. Additionally, our method reconstructs more consistent details in the doll's arm, as highlighted in red areas, while other video-based methods result in texture blurring issue.
>
> As illustrated in Figure. 6, taking complex and diverse scenario images as input, our method can generate better mesh and textures.
>
>
>
> Q3: Geometry consistency comparisons with natural images as input.
>
> A3: As mentioned in lines 264-269, the GSO dataset we used for evaluation is sampled from real-world objects, and results in our experiment are generated from natural images.
>
> Meanwhile, we have included complex results with different input types in Figure. 6 and Figure. 7 in the Appendix of our updated paper, such as from Objaverse image, Artistic image, Photographic image and Real-world captured image. As shown in Figure. 6, as highlighted in red areas, compared with SV3D and V3D, better geometry consistency can be obtained by our method.
>
>
>
> Q4: More ablation studies on the IVI module.
>
> A4: As mentioned in lines 453-470 of our main paper, we conducted both quantitative and qualitative ablation studies on the IVI module, including view interpolation for LRM (Figure. 5), camera trajectories (Figure. 4 and Table. 4), and interpolation number n (Table. 5).
>
> Besides, to fully evaluate the quantitative results, we also add the baseline (Wonder3D) results in Table. 4 of appendix in the updated version of our paper. The baseline results for Chamfer Dist/Vol. IoU/PSNR/LPIPS are 0.0186, 0.4398, 13.31 and 0.2554, while results with our IVI module are 0.0101, 0.6353, 18.27 and 0.1397 which outperform the baseline results with large margins on all evaluation metrics, which proves that our IVI module works positively for dense image generation. Please see Table. 4 for more details. Please kindly note that the baseline results are obtained with wonder3D without IVI module. Meanwhile, our IVI module with elevation can further improve the results.
>
> Furthermore, in Figure. 7 of the Appendix, we also provide more visual ablation study result with and without elevation. As shown in Figure. 7, with elevation in camera trajectories, our IVI module provides more details for video results, showing better quality in the reconstructed mesh, which is highlighted in red areas. For example, the fork of the forklift and the eyes of the dragon are completer and more refined.

---

> ### Comment · Reviewer_he5a · 2024-11-24
>
> Thank you for your rebuttal. I am wondering why are you not providing video results?

---

> > ### Author Response · Authors · 2024-11-25
> > **Re-down our main paper and supplementary materials.**
> >
> > Thank you for your response. We apology that we may mis-understand the response. We provide two kinds of video results.
> >
> > 1. We submit a new version of our main paper and the video results are in the appendix (Figure 6 and Figure 7). Please re-download our paper for more details.
> >
> > 2. We submit a new version of supplementary materials, which contains the rendered video comparison results between our approach and SOTAs, please re-download the supplementary materials for more details.

---

> > > ### Comment · Reviewer_he5a · 2024-11-25
> > >
> > > Thank you. I have read the videos. Although the video results are better than baselines, the results of baselines seems much worse than the results in their papers?

---

> > > > ### Author Response · Authors · 2024-11-26
> > > >
> > > > Thank you for your feedback. We appreciate the deep observations about the video results, and we agree that SV3D and V3D can generate reasonable results in general cases, such as the doll case in the video. However, due to lacking frame-to-frame consistency, geometry distortions always exist, which limits their performances.
> > > >
> > > > 1. We carefully reproduced the experimental settings of SV3D and V3D and ensured consistency in the experimental setup. SV3D, V3D and our approach are all trained on Objaverse dataset.
> > > >
> > > > 2. Both SV3D and V3D adopt a one-shot multi-frame generation approach for multi-view synthesis, which struggles to maintain frame-to-frame consistency. This limitation significantly impacts their ability to handle different types of cases.
> > > >
> > > >     For example, as shown in the video, for the doll case, the results of SV3D and V3D are reasonable, however, due to lacking frame-to-frame consistency, geometric distortions appear in finer details, such as the doll's arms. As shown in Figure. 3 of our main paper, the clock and rabbit results obtained by SV3D and V3D are also reasonable, however, geometry distortions also appear, such as the back of the rabbit and clock. Please re-download our main paper for more details.
> > > >
> > > >     Meanwhile, for cases with complex geometries, such as the forklift case, for SV3D and V3D, lacking frame-to-frame consistency leads to more serious geometric distortions, such as the forklift‘s arms and tires.
> > > >
> > > > Our designed IVI module can bring better multi-view consistency, and results obtained by our approach are with better geometry and textures.

---

### Official Review · Reviewer_D6WW · 2024-11-04

**Soundness:** 2
**Presentation:** 2
**Contribution:** 2
**Rating:** 5
**Confidence:** 4

**Summary:**

This paper introduces a pipeline for reconstructing or generating 3D textured object meshes from a single image. The pipeline is built on the recent framework of multi-view diffusion combined with large reconstruction models (LRMs). If I understand correctly, the central claim of this paper is that "interpolated view synthesis using image diffusion models can provide more information and enhance 3D consistency and lead to improved object reconstruction results in LRMs." The motivation, as outlined in the paper, is that (1) Sparse image-to-multi-view models are not enough for LRM models, and (2) previous dense image-to-multi-view diffusion models (like video models) lack pixel-accurate 3D consistency, which can negatively impact the reconstruction quality of LRM models.

The authors specifically propose an infinite-view generation model that is conditioned on continuous camera poses rather than discrete ones. The images generated from these interpolated camera poses are then used as inputs to the LRM model to create a 3D mesh.

Overall, I find the paper's main points and motivations challenging to follow. From my experience, using dense views as input images does not benefit LRM models. Given this, it's unclear to me how interpolating between these inconsistent views would improve 3D consistency. I struggle to see how this approach addresses the core issue.

**Strengths:**

- This paper is easy to read with clean writing.
- The problem definition is clear.
- The paper compares the recent state-of-the-art image to 3D models, which is good.

**Weaknesses:**

- As stated in the summary section, I do not think for the object reconstruction task, adding dense views for LRM models will help the reconstruction.
- The paper lacks in video results, or dense image results to show the quality of this “infinite” multi-view diffusion model. This part is crucial to help readers appreciate the whole method.
- The paper lacks quantitative ablation study on the view-interpolation design. There is only a qualitative comparison in Fig. 5, which is not convincing enough. Actually, I suspect the rationale of such multi-view interpolation design.

**Questions:**

- In equation 3, where $n$ is defined?
- As the number of input images increases for the LRM model, how does the time required for training and inference compare to that of the sparse-view version of the LRM model?
- Also, as the number of output images increase for the multi-view generation model,  how does the time required for training and inference compare to that of the sparse-view version of multi-view generation model?

---

> ### Author Response · Authors · 2024-11-20
>
> Thank you for appreciating the clean writing, clear problem definition, and our comparison with recent state-of-the-art models. Your feedback is invaluable, and we'll work on addressing the concerns you've raised to improve the clarity and impact of our work.
>
> Please kindly note that we update more experiment results and analysis in our updated paper, please re-download our main paper and supplementary materials for more details.
>
> Q1: Adding dense views for LRM models will help the reconstruction.
>
> A1: In this paper, we argue that dense views with better inter-frame consistencies help the reconstruction, not only dense views. As mentioned in lines 42-43 and 54-70 of our main paper, simply adding dense views does not ensure better reconstruction results. Due to lacking inter-frame consistency, the results of video-based approaches, such as SV3D and V3D, in Table. 2, Figure. 1 and Figure. 3 of our main paper can obtain limited performances.
>
> As mentioned in lines 72-77 and 216-219 of our main paper, our IVI module utilizes two main views as condition and reference for view interpolation, which achieves better multi-view consistency, compared to video-based methods like SV3D and V3D, leading to superior reconstruction results.
>
> We provide more results in Appendix of our updated paper, which proves the effectiveness of our approach. As shown in the results in Figure 6, better multi-view consistency images can be obtained by our approach, compared with other video-based methods. For example, our method outperforms other video-based methods with more accurate geometry details in the forklift, cat, milk bottle and doll's arm.
>
> Please kindly note that we provide results with different types of images as input, such as images from real-world, etc., and our model is only trained with objaverse dataset, which proves the generalization ability of our approach.
>
>
>
> Q2: More video results and dense image results to show the quality of this “infinite” multi-view diffusion model.
>
> A2: Thank you for pointing this out. We have added video results in Figure. 6 and Section. B in the Appendix of our updated paper.
>
> As shown in the results in Figure. 6 of the Appendix, better multi-view consistency images can be obtained by our approach, compared with other video-based methods, and differences in the mesh results are highlighted in red areas. For example, our method outperforms other video-based methods with more accurate geometry details in the forklift and cat, while SV3D and V3D show flattened results, treating three-dimensional objects as nearly two-dimensional objects. In the milk case, our approach effectively converts 2D artistic images into consistent multi-view images and intact meshes, maintaining shape consistency that others fail to achieve. Additionally, our method reconstructs more consistent details in the doll's arm, as highlighted in red areas, while other video-based methods result in texture blurring issue.
>
>
> Q3: Quantitative ablation study on the view-interpolation design
>
> A3: Thank you for pointing this out. As mentioned in lines 462-465 and 467-470 of our main paper, we conducted quantitative analysis on camera trajectories in Table. 4 and interpolation number n in Table. 5.
>
> Besides, we also add the baseline (Wonder3D) results in Table. 4 in the appendix of the updated version of our paper. The baseline results are obtained with wonder3D without IVI module. As shown in Table. 4, results with our IVI module outperforms the baseline without IVI with large margins on all evaluation metrics, especially for "Chamfer Dist", "Vol. IoU", "PSNR" and "LPIPS", which proves that our IVI module works positively for dense image generation.
>
>
>
> Q4. The definition of n in equation 3.
>
> A4: Thank you for pointing this out. n represents the number of interpolated images during a single view interpolation process. We update the description in line 214 of our revised paper.
>
>
>
> Q5. Training and inference time for different number of input images for LRM model.
>
> A5:  Training time and Inference time for mesh reconstruction:  Our 3D mesh reconstruction LRM part takes about 1 day for training on 8 Nvidia A100 GPUs and an average time of 1.464 seconds for inference, which is similar with InstantMesh that constructs meshes in an average time of 1.270 seconds.
>
> As shown in Figure 2 (c) and Equation 4 of our main paper, all image tokens are concatenated for subsequent operations. We have a position embedding p∈R^{V,P,D} and a concatenated tensor X∈R^{V,P,D}, where V represents the number of view images. p serves as the query and X acts as the key in the cross-modal attention operation. The matrix multiplication is mainly performed along P and D dimensions. In our experiment, P is 401 and D is 768 which are significantly larger than V. Therefore, increasing V has minimal impact on computational time.

---

> > ### Comment · Reviewer_D6WW · 2024-11-27
> > **Sorry but one more question**
> >
> > Sorry but one more question, I can only find video results of rendering of reconstructed meshes. What is the visual consistency like on the video results using images from interpolated view synthesis module? (i.e., the IVI module).

---

> > > ### Author Response · Authors · 2024-11-27
> > >
> > > Thank you for your feedback.
> > >
> > > 1.	We appreciate the deep considerations about the training/inferencing time.
> > >
> > > (1). Please kindly note that our approach does not result in a computational time proportional to $V^2$. Our average inference time is 1.464 seconds, which is similar with InstantMesh that constructs meshes in an average time of 1.270 seconds. This is because we only increase the computational load in the image encoder's transformer (cross-modal attention) part. After this step, we employ a Triplane transformer that concatenates and flattens features from all views, then decodes them into a fixed-shape Triplane. Subsequent operations are based on this fixed-shape Triplane, which does not increase computational overhead. Therefore, the additional computational time is primarily confined to the image encoder section, and the overall computational complexity is not proportional to $V^2$.
> > >
> > > (2). Besides, as we described before, for the concated tensor X∈R^{V,P,D}, as analyzed in [1], though the theoretical time complexity of cross attention is O((VP)^2, D), we use pytorch in our experiments, the matrix multiplication is mainly performed along P and D dimensions, and as described in the pytorch document, “FlashAttention-2: Faster Attention with Better Parallelism and Work Partitioning” and “Memory-Efficient Attention” are utilized to accelerate the attention process.
> > > Thus increase of V bring acceptable time consuming, from 1.270 seconds to 1.464 seconds.
> > >
> > > Thank you for your feedback, we have updated more details in the Section. A of the appendix of our paper.
> > >
> > > [1]. Vaswani, A. "Attention is all you need." Advances in Neural Information Processing Systems (2017).
> > >
> > >
> > > 2.	We submit a new version of supplementary materials and update the “video_results.mp4”. The generated video results of IVI module and results of SV3D and V3D are included.
> > >
> > > SV3D and V3D can generate reasonable results for general cases, however, as shown in the video, the inconsistencies commonly exist, such as the tires of forklift, the bowknot of the milk bottle, the legs and the arms of the doll. Please see the updated “video_results.mp4” for more details.

---

> ### Comment · Reviewer_D6WW · 2024-11-27
> **Response**
>
> Thank you authors for the detailed response.  I still have questions about your last answer. Isn't the training/inferencing time proportional to $V^2$ as you performed cross attention between $p \in \mathbb{R}^{V,P,D}$ and$ x \in \mathbb{R}^{V,P,D}$?

---

### Official Review · Reviewer_8fJs · 2024-11-04

**Soundness:** 3
**Presentation:** 3
**Contribution:** 3
**Rating:** 5
**Confidence:** 4

**Summary:**

The paper introduces InfiniteMesh, a framework for generating high-quality 3D meshes from a single image by combining multi-view diffusion-based view interpolation and mesh reconstruction techniques. The main task is to create consistent, high-fidelity 3D representations from limited initial views, addressing common issues like inconsistency and low detail in existing methods. Key contributions include the Infinite View Interpolation (IVI) module, which enhances multi-view consistency by generating interpolated views between initial main views, and a tri-plane-based reconstruction model that processes these views to produce accurate 3D meshes. Experimental results show that InfiniteMesh outperforms state-of-the-art approaches in both geometric and textural quality, especially on the Google Scanned Objects dataset.

**Strengths:**

1. The IVI module is a novel contribution that improves multi-view consistency by generating interpolated views between 4 main views generated by Wonder3D. This approach minimizes inconsistencies across views, addressing a common challenge in 3D mesh generation.

2. The combination of multi-view interpolation and a tri-plane-based reconstruction model results in superior mesh quality, with enhanced detail in both geometry and texture. The framework effectively captures intricate features and realistic textures, outperforming current state-of-the-art models both qualitatively and quantitatively.

**Weaknesses:**

1. Inference Time missing: Infinite Mesh reports stronger quantitative performance and slightly better geometry and texture synthesis than several baselines. However, I am concerned about the extra computation time incurred by the IVI module for synthesizing sufficient novel views for multiview consistency. Instant Mesh constructs a mesh in 10 seconds, I doubt whether this method can get close to such speeds given the IVI overhead. For a fair comparison, the authors should report and compare average mesh creation times with the baselines considered in the paper.
2. The ablation with the camera trajectories for the IVI module does not seem too convincing, both qualitatively and quantitatively. The NVS results for the 2 hand-picked novel views are mostly similar, and the meshes have similar geometry details.
3. More details on mesh reconstruction: The authors should ideally provide more details and the motivation behind specific loss functions for mesh reconstruction in Equation 7, either in the main paper or the supplementary. This would save the reader (unfamiliar with all related literature, like the RODIN paper) from spending too much time reading other papers to understand the method correctly.
4. No 360 reconstruction videos: The authors should ideally share 360 reconstruction videos in the supplementary so that the multiview consistency introduced by the LRM and IVI modules can be better evaluated. Renderings of a few hand-picked novel views do not do much to convince the reader. The IVI module is introduced as a more compute-friendly alternative to video diffusion models to ensure multiview consistency. However, current qualitative results do not support that entirely.

**[Update post rebuttal]**: I would like to thank the authors for their efforts during the rebuttal phase and for responding to the reviewers' concerns earnestly.

- Some of my doubts and questions were answered in the rebuttal, while others remain. For example, I am still unsure about the total number of main and interpolated views used for mesh reconstruction. An exact number can be easily cited.

 - Additional qualitative results in the supplementary show the benefits of the IVI module for multiview consistency, but still, most of the images (real-world/photographic) are inclined towards a synthetic setting. The author's reluctance to share reconstructions with more challenging real-world images as in competing methods weakens the experimental evaluation. For a more convincing comparison, the authors could easily have picked images used in works like Syncdreamer / Wonder3D rather than having a custom evaluation protocol.

- I would like to thank Reviewer Ystn for pointing out the discrepancy in previously reported performance of competing methods with the numbers reported in Table 2. I do not find the author's response satisfactory in changing the mesh texture evaluation protocol, whereas methods like Wonder3D report the performance of existing works as is. By following established experimental settings, this confusion could have been easily avoided.

- I would encourage the authors to tweak some of their experiments for more convincing comparisons and resubmit to a future conference as I believe the proposed IVI module has merits. But for now, I would have to decrease my score.

**Questions:**

1. How many interpolated views are used in total to optimize the mesh reconstruction? From what I understand, Wonder3D initially produces 4 views from a single image, and then the IVI module generates n = 2 interpolated views each on left and right for these 4 main views - so in total, 20 views are used for reconstruction (16 interpolated and 4 from Wonder3D) Is that correct?

---

> ### Author Response · Authors · 2024-11-20
>
> Thank you for appreciating the contributions of our IVI module in improving multi-view consistency and mesh quality. Your acknowledgment of our work outperforming current state-of-the-art models is greatly appreciated.
>
> Please kindly note that we update more experiment results and analysis in our updated paper, please re-download our main paper and supplementary materials for more details.
>
> Q1: Inference time comparisons of mesh reconstruction and view interpolation
>
> A1: (1). Mesh reconstruction: Our 3D mesh reconstruction LRM part takes an average time of 1.464 seconds for inference, which is similar with InstantMesh that constructs meshes in an average time of 1.270 seconds.
>
> As shown in Figure 2 (c) and Equation 4 of our main paper, all image tokens are concatenated for subsequent operations. We have a position embedding p∈R^{V,P,D} and a concatenated tensor X∈R^{V,P,D}, where V represents the number of view images. p serves as the query and X acts as the key in the cross-modal attention operation. The matrix multiplication is mainly performed along P and D dimensions. In our experiment, P is 401 and D is 768, which are significantly larger than V. Therefore, increasing V has minimal impact on computational time.
>
> (2). View Interpolation: Our IVI module takes 3.5s for a single view interpolation process. In our experiment, four interpolations are required, the total video generation time is approximately 14s.
>
> The quantitative comparison results with SOTA video generation methods are as follows:
>
> SV3D:           85.198 seconds
>
> V3D:          31.893 seconds
>
> IVI (Ours):    14.324 seconds
>
> Compared with SV3D and V3D, our View Interpolation module (IVI) is with less time consumption. Please kindly note that all results are obtained with a A40 GPU.
>
>
> Q2: Ablation study of IVI module's camera trajectories
>
> A2: Thank you for this observation.  As shown in Figure. 4 of our main paper, taking a rabbit image as input, compared with IVI module without elevation, results of our IVI module with elevation in camera trajectories can reconstruct better mesh with better back and hand (red areas in Figure. 4 of our main paper).
>
> Meanwhile, we add more qualitative results on camera trajectories with more distinctive examples in Figure. 7 of Appendix of our updated main paper. With elevation in camera trajectories, results with our IVI module show better quality in the reconstructed mesh, which is highlighted in red areas. For example, the fork of the forklift and the eyes of the dragon are completer and more refined.
>
> Besides, In Table. 4 of our main paper, we provide quantitative results of our method with different elevation angles (camera trajectories). We also add the baseline (Wonder3D) results in Table. 4 of the updated version of our paper. The baseline results are obtained with wonder3D since we use it as baseline without IVI module. As shown in Table. 4, results with our designed camera trajectories with and without elevation all outperform baseline with large margins, which proves that all our designed camera trajectories work positively for dense image generation, and trajectories with elevation can further improve the performances on all evaluation metrics. This is because that most of the areas can be observed in the trajectory without elevation, and the rarely observed areas can be observed by trajectories with elevation.
>
>
> Q3: Explanation of loss functions in Equation 7
>
> A3: Thank you for highlighting this. In Equation 7, Lrgb, Ldepth, Lnormal, and Lmask refer to the loss of RGB images, depth, normal, and mask maps of the reconstructed mesh, and Llpips and Lreg refer to LPIPS and regression loss, respectively. We have added this in E of Appendix of our updated paper.
>
>
> Q4: 360-degree reconstruction videos
>
> A4: Thank you for this valuable suggestion. We provide more visual results in Figure. 6 in Appendix of our updated paper with different types of input. We also provide more 360° reconstruction dense images in Figure. 7 of the Appendix.
>
> As shown in the results in Figure 6, better multi-view consistency images can be obtained by our approach, compared with other video-based methods. For example, our method outperforms other video-based methods with more accurate geometry details in the forklift, cat, milk bottle and doll's arm.
>
> Our method outperforms SV3D and V3D with better mesh and texture reconstruction on different types of input, such as such as images from real-world, etc. Please kindly note that our model is only trained with objaverse dataset.
>
>
> Q5: Interpolated views for mesh reconstruction
>
> A5:  We generate n=2 interpolated views between each pair of adjacent main views.  As shown in Table 5 of our main paper, better results on most of the evaluation evaluation metrics can be obtained when set n=2, therefore, we set n=2 in our experiment.

---

### Meta-Review · Area_Chair_MCjp · 2024-12-17

**Metareview:**

This paper presents InfiniteMesh to generate 3D textured meshes from a single image. An IVI module is proposed to generate interpolated views so that it enhances the 3D consistency for the following tri-plane-based mesh reconstruction step. The 3D inconsistency in multi-view diffusion is challenging but important for image-to-3D tasks, and the quality of the generated meshes is good. However, there are some concerns raised by the reviewers and still remain after rebuttal. The experiments should be further revised, including a clearer explanation of the comparison with previous works as there are significant discrepancies in the reported PSNR, missing details in experimental settings, evaluations on challenging real-world images instead of the simple cases, etc.. Also, the novelty of the key module IVI should be further highlighted as it is somehow engineering and makes the whole pipeline more like a combination of existing methods. Due to the concerns, I recommend a decision of rejection of this paper.

**Additional Comments On Reviewer Discussion:**

Initially the reviewers have concerns on experiments, novelty and some technical/experimental details of the paper. In the rebuttal, the authors address some of them, but there are still important issues remaining as I summarize in the metareview, which should be well-addressed before acceptance.

---

### Decision · Program_Chairs · 2025-01-22

Reject